# Antimalarial mass drug administration in large populations and the evolution of drug resistance

Tran Dang Nguyen [1⦿], Thu Nguyen-Anh Tran [1⦿], Daniel M. Parker [2], Nicholas J. White [3,4], Maciej F. Boni [1,3]*

**1** Center for Infectious Disease Dynamics, Department of Biology, Pennsylvania State University, PA, United States of America, **2** Department of Population Health and Disease Prevention, Department of Epidemiology and Biostatistics, University of California, Irvine, Irvine, CA, United States of America, **3** Centre for Tropical Medicine and Global Health, Nuffield Department of Medicine, University of Oxford, Oxford, United Kingdom, **4** Mahidol-Oxford Research Unit, Wellcome Trust Major Overseas Programme, Mahidol University, Bangkok, Thailand

⦿ These authors contributed equally to this work.

\* mfb9@psu.edu

**Data Availability Statement:** All simulation code, simulation output, and MDA participation data are publicly available at https://github.com/bonilab/malariaibm-MDA-2018WHOERG.

## Abstract

Mass drug administration (MDA) with antimalarials has been shown to reduce prevalence and interrupt transmission in small populations, in populations with reliable access to anti-malarial drugs, and in populations where sustained improvements in diagnosis and treatment are possible. In addition, when MDA is effective it eliminates both drug-resistant parasites and drug-sensitive parasites, which has the long-term benefit of extending the useful therapeutic life of first-line therapies for all populations, not just the focal population where MDA was carried out. However, in order to plan elimination measures effectively, it is necessary to characterize the conditions under which failed MDA could exacerbate resistance. We use an individual-based stochastic model of *Plasmodium falciparum* transmission to evaluate this risk for MDA using dihydroartemisinin-piperaquine (DHA-PPQ), in populations where access to antimalarial treatments may not be uniformly high and where re-importation of drug-resistant parasites may be common. We find that artemisinin-resistance evolution at the *kelch13* locus can be accelerated by MDA when all three of the following conditions are met: (1) strong genetic bottlenecking that falls short of elimination, (2) re-importation of artemisinin-resistant genotypes, and (3) continued selection pressure during routine case management post-MDA. Accelerated resistance levels are not immediate but follow the rebound of malaria cases post-MDA, if this is allowed to occur. Crucially, resistance is driven by the selection pressure during routine case management post-MDA and not the selection pressure exerted during the MDA itself. Second, we find that increasing treatment coverage post-MDA increases the probability of local elimination in low-transmission regions (prevalence < 2%) in scenarios with both low and high levels of drug-resistance importation. This emphasizes the importance of planning for and supporting high coverage of diagnosis and treatment post-MDA.

**Funding:** This work was commissioned for WHO Evidence Review Group (ERG) on Mass Drug Administration for Malaria (Sep 11-13, 2018, Geneva). The work was funded by the Malaria Modeling Consortium (www. malariamodelingconsortium.org) via the University of Washington's MMC grant from the Bill and Melinda Gates Foundation (OPP159934) to MB. Model simulations were done at Penn State's Institute for Computational Data Sciences (ICDS) Advance Cyberinfrastructure (ACI) computing cluster. DMP is supported by funds from the Bill and Melinda Gates Foundation (OPP1211806) and the National Institutes of Allergy and Infectious Disease (FAIN: U19AI089672; Subaward number: 6123-1247-00-D). The funders had no role in study design, data collection and analysis, decision to publish, or preparation of the manuscript.

**Competing interests:** The authors have declared that no competing interests exist.

# 1 Introduction

Global malaria burden dropped substantially between 2000 and 2015, prompting the World Health Organization to once again articulate the case for and evaluate the feasibility of global malaria eradication [1, 2]. These efforts have been progressing via national-level malaria elimination plans in countries that are nearing elimination phase, and mass drug administration (MDA) will be one of the key public health tools deployed in regions or transmission pockets where complete elimination has otherwise proved difficult. In small populations and populations with low malaria prevalence, MDA implementations have proved successful at pushing infection numbers to zero [3, 4]. In larger populations or certain high-prevalence scenarios, MDA is often followed by a rebound of malaria cases after the intended rounds of MDA have completed. Rebounds of case numbers and malaria prevalence have been observed in the field months or years after the transmission-reducing effects of the MDA have worn off [5–11], an effect that is predicted by mathematical modeling analyses [12–14] if malaria control efforts are not intensified from the pre-MDA status quo. An important question in MDA application and follow-up is how to keep transmission levels low with the eventual goal of interrupting transmission permanently.

A second concern addressed here is the natural selection of drug-resistant *Plasmodium falciparum* genotypes during MDA [15, 16]. Ideally, a therapy chosen for MDA would have high efficacy in patients and no prior history of use in the focal population, in order to minimize the probability of pre-existing drug resistance being selected for during the MDA. These measures may mitigate the risk, but the MDA's core approach–to dose every resident in a population with antimalarial drugs whether parasitaemic or not–is necessarily accompanied by short-lived but substantial selection pressure for drug resistance. Quantifying this risk is not straightforward as resistance frequencies can go up while prevalence is dropping, which is why it is critical to combine drug-resistance risks (net negatives) with likelihood of malaria elimination (a net positive) into a single unified analysis. In some modeled scenarios, drug-resistant alleles can appear to fix while the parasite population is on the path to elimination. An evaluation of an MDA's benefits and risks needs to account for drug-resistant allele frequencies, absolute numbers of individuals carrying resistant genotypes, the probability that elimination will occur before resistant genotypes fix in the target population, and the long-term number of treatment failures experienced in the population.

In small populations and low-prevalence scenarios, the balance of field experience indicates that MDA leads to net positive outcomes with high likelihood of local elimination and little drug resistance risk [3, 4], despite the difficulty of collecting randomized controlled data from these large studies [17]. In this study, we evaluate drug-resistance risks associated with MDA programs deployed in large populations (>40K individuals, as determined by a WHO working group [18]) where prevalence may still be moderate to high and ACT adoption may be low; these scenarios are motivated by African epidemiological contexts where diagnosis, treatment, and ACT adoption are all lower than in Southeast Asia. Using a previously developed individual-based mathematical transmission model of *Plasmodium falciparum* [19–21] we explicitly model the small numbers of parasite positive individuals post-MDA and describe the conditions under which drift, selection, and migration can act synergistically to accelerate the fixation of drug-resistant genotypes. Finally, we outline epidemiological scenarios where MDA policies are likely to pose the most and least risk to near-term and long-term drug-resistance trends.

# 2 Methods

## 2.1 Model description

We modified a previously published individual-based mathematical model of *P. falciparum* transmission and evolution [19–21]. The model includes many standard epidemiological and

clinical features of malaria transmission, summarized in S1 Text. One major new upgrade in the current model version is a locus-based resistance framework, with drug-resistance phenotypes parameterized for the K76T locus (*pfcrt* gene), N86Y and Y184F loci (*pfmdr1* gene), C580Y locus (*kelch13* gene), copy number of *pfmdr1*, and a composite locus corresponding to the piperaquine-resistant phenotypes observed in Southeast Asia last decade [22, 23]. Drug resistance of a *P. falciparum* genotype in the simulation is fully described by these six genetic features. Each genotype-drug combination ($64 \times 6 = 384$ combinations) has assigned to it its own parasite killing rate value ($p_{max}$) and its own EC50 value (the drug concentration at which the killing rate is $p_{max}$ / 2) from which we derive the efficacy of each therapy/drug on each parasite genotype; see S2 Text for the parametrization of the drug-by-genotype table used in these analyses. As the science around piperaquine (PPQ) resistance has been developing rapidly over the past five years, the composite PPQ-resistance locus in the simulation can be viewed as a combined genotype possessing certain *pfcrt* mutations (most likely candidates being T93S, H97Y, F145I, I218S, and G353V) that are known to show moderate to strong in vitro resistance in piperaquine resistance assays [24–30].

In addition, each resistant genotype is assigned a daily fitness cost $c_R = 1 - (1 - 0.0005)^n$ as in previous modeling exercises [19–21] that increases with the number $n$ of resistant alleles or copy-number variants. This corresponds to a 17% annual fitness cost in terms of within-host parasitaemia level, and thus has an effect on gametocyte production (in our model) and onward transmission to mosquitoes. This fitness cost is within an order of magnitude of the multi-year timelines that have been observed for drug-resistant alleles receding from a population when a drug is withdrawn from use.

Individuals newly infected by *P. falciparum* may experience symptoms (depending on their immune status), and 55% of symptomatic individuals (60% if <5) will seek and receive antimalarial treatment. At the beginning of the simulation it is assumed that among treatment-seeking individuals, 36% will receive artemether-lumefantrine (AL) [31] which is the first-line policy in our modeled scenario, and the remaining 64% will purchase either SP (assumed efficacy of 40%; but see Figs I to L in S1 Text), amodiaquine, chloroquine, or AL on the private market; these are typical values in many African settings. Assuming that ACT adoption continues to increase, we set an annual increase of 2% per year in ACT use, and the model's 36% public-market use in year zero increases to 76% twenty years later [31].

A second major model upgrade allows for mass drug administration (MDA). Each individual in the population is assigned a probability of participating in a single round of MDA, as some individuals by the nature of their work, travel, or age will be less likely or more likely to be present when MDA team members are present and distributing treatments [32, 33]. Each individual's participation probability is drawn from a beta distribution with mean 0.75 for individuals aged 10–40, and mean 0.85 [34] for individuals aged <10 or >40; standard deviation of this beta distribution is set at 0.3 [35, 36] for all ages.

As *P. falciparum* population sizes after the MDA may be quite low, we evaluate the hypothesis that importation rather than mutation presents the primary drug-resistance risk. Importation is modeled as one new parasite coming into the population every 10 days; different rates of importation are noted where appropriate. Thus, our standard importation scenario is intended to model a region or site where re-importation of drug resistance would be common and expected, in order to evaluate a situation perceived as having maximum risk. The genotype of the imported parasite is chosen at random with equal probabilities for alleles and copy numbers for all loci. This means that during an MDA, an imported parasite has a 25% chance of being a double-resistant to both DHA and PPQ and a 50% chance of being resistant to exactly one of the drugs in the combination. Mutations do arise *de novo* in the blood stage [37] in the simulation, at all loci, only if the mutated phenotype carries an advantage in the present

within-host drug environment (by being associated with lower drug efficacy). In a small minority of simulations, 580Y alleles may exist at low frequencies prior to the start of an MDA campaign due to early *de novo* mutation.

As the effects of stochasticity are central and influential in the model runs, we carried out an evaluation comparing outcomes from $N = 1000$ simulations to $N = 100$ simulations. None of the major conclusions on drug resistance risk changed from $N = 100$ to $N = 1000$ simulation runs per scenario, thus most analyses are presented as quantiles and medians from $N = 100$ simulation runs. Sections 7 and 8 in S1 Text describe the full range of outcomes and outline some differences among the quantile distributions from $N = 100$ and $N = 1000$ simulations.

## 2.2 Outcome measures

Allele frequency is computed using a weighting approach to account for individuals with multi-clonal falciparum infections. The frequency of an allele or genotype is computed as the weighted number of individuals carrying that allele or genotype divided by the total number of parasite-positive individuals. The weight (between zero and one) of an individual in this sum is their number of parasite clones carrying the allele divided by the person's total number of clones circulating in the blood.

The *kelch13* 580Y allele frequency is used as a proxy for artemisinin-resistant phenotypes in our analysis. The *P. falciparum* blood-slide prevalence in 2–10 year-olds (PfPR$_{2-10}$ [38]) is used to evaluate the success of MDA and any subsequent improved treatment coverage (ITC) after the MDA has completed. The time it takes for 580Y allele frequencies to reach 0.25 (we use the notation $T_{0.25}$) is used as a measure of how quickly artemisinin resistance reaches an irreversible course of fixation; this is a milestone at which it is too late to reverse the spread of resistance. Differences among distributions are presented using Kruskal-Wallis (KW) tests, treating multiple simulation runs as independent samples from the set of all evolutionary-epidemiological outcomes that are possible given a particular set of fixed assumptions. Note that KW tests, like all nonparametric tests, can have $p$-values below the $10^{-100}$ range when sample sizes are large ($N = 1000$) so the effect sizes must be examined to determine if differences among groups are meaningful.

## 2.3 Scenarios modeled

For this modeling exercise, populations of 40,000 and 300,000 are considered [18], and the planned MDA targets all individuals in the population, although only 75% to 85% will be present for participation for any given round of MDA. Four rounds of MDA are administered five weeks apart; it takes two weeks to complete one round and the campaign is completed in 17 weeks. The start date for the first round is January 1, and the subsequent rounds start on February 5, March 12, and April 16. The antimalarial used in each round of MDA is a three-day course of DHA-PPQ. Simulations are run for twenty years to investigate long-term patterns of drug resistance following a single MDA program run in the first year only. Sensitivity analyses on the population size and number of MDA rounds are conducted.

As it has been raised in a number of MDA trials that improved coverage and increased treatment access are key to lowering prevalence and incidence [3, 6, 18, 33, 39], we implemented a model feature where improved treatment coverage (ITC) follows the MDA. In this scenario, the program's infrastructure and improved knowledge on antimalarial use is leveraged to increase treatment coverage from 55%/60% (pre-MDA levels) to 80% of all symptomatic malaria cases treated (unless another percentage is noted); the increase occurs linearly over a six-month period and remains constant for the remainder of the simulation. *P. falciparum* prevalence levels investigated ranged from 1% to 5%; this range was selected as higher

prevalence scenarios (10%, 15%) did not lead to elimination and had qualitatively similar dynamics to a 5% setting; this was a part of our initial scenario selection process to ensure we chose scenarios where MDA would be considered but would not be guaranteed to be immediately successful. Initial genotypes in the population are 50% N86 and 50% 86Y at the N86Y locus of *pfmdr1*. All other loci are *pfcrt*-76T, *pfmdr1*-Y184 single copy, *kelch13*-C580, and piperaquine sensitivity at the relevant *pfcrt* loci; variation at the N86Y locus only is included due to the multitude of effects this locus has on resistance phenotypes to different drugs (S2 Text). A sensitivity analysis using Latin hypercube sampling on malaria prevalence, importation, MDA coverage, post-MDA treatment coverage, number of MDA rounds, the drug-resistance fitness cost, and population size was conducted, and partial rank correlation coefficients (PRCC) with $T_{0.25}$ are presented (S2 Text, section 5).

## 3 Results

### 3.1 Effects of MDA on drug-resistance evolution

Implementation of high-coverage mass drug administration is followed by a drop in prevalence which results in a genetic bottleneck lasting months or years. When drug-resistance alleles are present or imported, whether at low or moderate frequencies, this bottleneck period is marked by substantial uncertainty and the near-term evolutionary outcome post-MDA will have low predictability. This is an unavoidable outcome of the action of random genetic drift in small populations [40–42] as successful MDA will necessarily reduce parasite population size and genetic diversity; we call this period a "resistance bottleneck" when drug-resistant alleles are present at high frequencies (Fig 1). Traditionally, the population genetics literature treats a bottleneck as a genotype-neutral event in which the expected reduction in population size for each allele is the same (i.e. no allele is preferentially selected), and the stochastic nature of the intervention results in randomly altered allele frequencies when comparing allelic distributions pre- and post-bottleneck. Under an MDA program there is one important difference, namely, that the cause of the population crash is non-neutral in its effects on the standing genetic variation in the parasite population as the MDA itself selects for drug-resistant alleles. Thus, an MDA can be viewed as a bottleneck created by negative viability selection. In the following analysis, we present both the population biological effects of MDA (reductions in malaria prevalence) and the population-genetic effects (changes in genotype distribution) resulting from the bottleneck following MDA.

When population sizes are low (<10,000 individuals) or when prevalence levels are low (PfPR < 1%), MDA leads either to elimination or low-level persistence with dozens of infected individuals present (Fig 1). The simple reason is that populations starting with several hundred *Plasmodium* carriers will have these numbers reduced to dozens after the MDA, numbers that are low enough that they may lead to local elimination or a stuttering chain of transmission for a prolonged period. Rapid rebounds are rare in these scenarios. In modeling larger populations (>40K individuals), epidemiological rebounds do occur after an MDA (Fig 2A and 2D), and in these scenarios the basic reproduction ratio of *P. falciparum* will bring prevalence back to an equilibrium which will be slightly lower than the pre-MDA equilibrium level. As we assume that ACT adoption is increasing slowly over the twenty years of the simulation, equilibrium prevalence will be lower at the end of the simulation than at the beginning. However, increasing ACT adoption is not rapid enough to interrupt malaria transmission permanently in the months following the last round of mass drug administration.

In a baseline scenario with 40,000 individuals, no parasite importation, and $PfPR_{2-10}$ = 2% (Fig 2A), one round of MDA brings median prevalence down to 0.37% after four months followed by a rebound in prevalence to 1.88% five years later; four rounds of MDA results in a

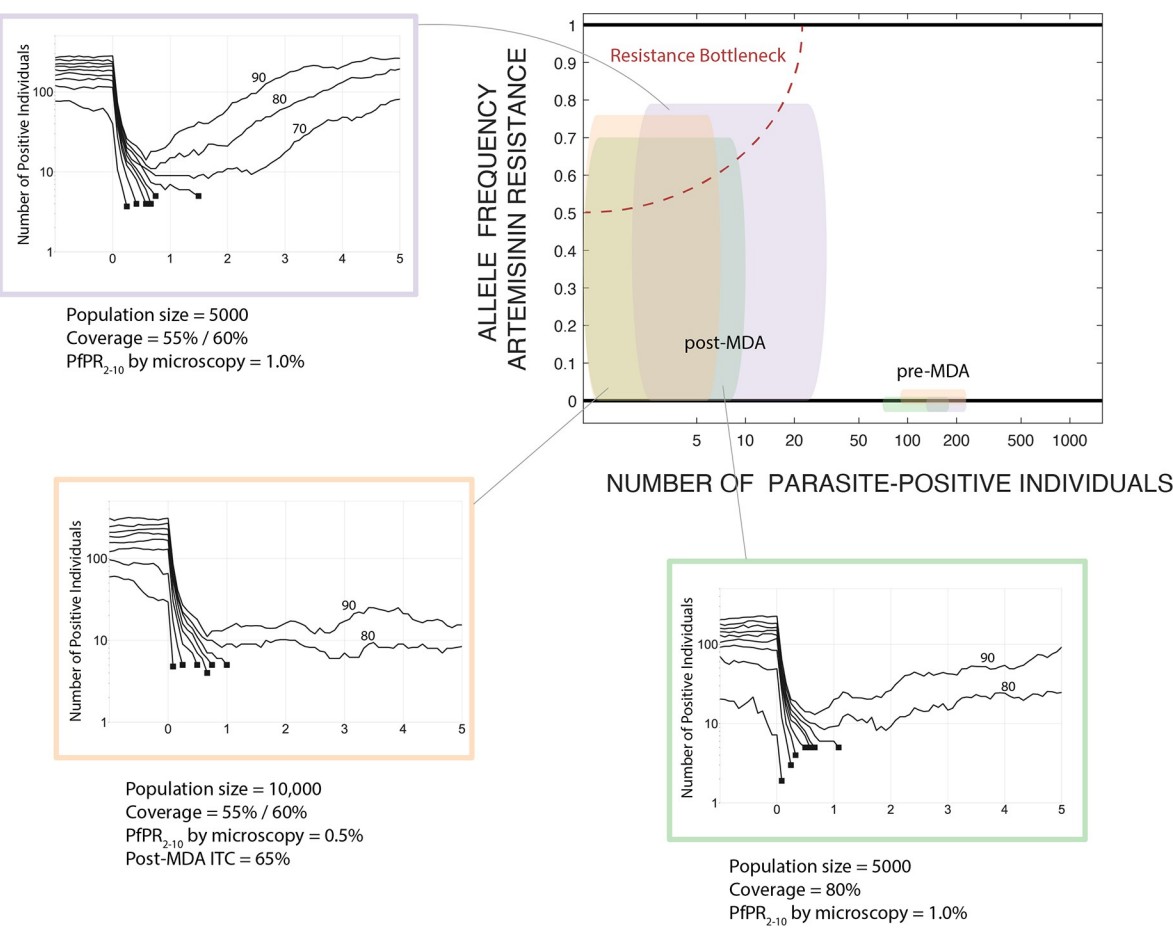

**Fig 1. Schematic showing the relationship between prevalence and drug-resistance pre- and post-MDA, and three small-population example scenarios showing the effects of MDA on parasite population size and probability of elimination.** The three scenario plots summarize 100 simulations under three rounds of MDA by showing the deciles (10th percentile to 90th percentile) of the simulation outcomes, with the non-extinct trajectories labelled by their decile. The *y*-axis shows the number of parasite-positive individuals including those that are not detectable by microscopy; *x*-axis shows five years. Simulated trajectories that reach extinction/elimination (<5 parasite-positive individuals) end in a black square and are not continued. In the orange box, the first decile is not shown as the number of parasite-positive individuals was <5 prior to the MDA. All examples include importation of drug-resistance of approximately two artemisinin-resistant genotypes per year. Elimination occurs for ~70% of simulations in the scenarios shown, and for ~80% of scenarios the number of parasite-positive individuals stays below 20 for at least three years. These example scenarios are chosen as boundary examples; in all three, probability of elimination increases with an extra round of MDA, lower importation rates of drug-resistance, lower pre-MDA prevalence, higher treatment coverage, or improved treatment coverage (ITC) post-MDA. The top-right panel shows the change in the parasite population profile from pre- to post-MDA. The box widths and heights show the inter-quartile range for the number of parasite-positive individuals (any level of parasitaemia) and the frequency of 580Y alleles, respectively, with colors matching the three example scenarios. The way that resistance risk is introduced into MDA scenarios is that the parasite population is pushed into this graph's upper-left corner, the resistance bottleneck. In the example simulations corresponding to the colored boxes, the parasite population sizes drop to the single digits or low double digits, but 580Y allele frequencies are unpredictable and can potentially be high, making the future path of drug resistance highly unpredictable.

median prevalence of 0.019% (a total of 16 infected individuals at its lowest point) with the median simulation rebounding to 1.24% prevalence. When four rounds of MDA were carried out, 440/1000 simulations resulted in malaria elimination; elimination was also observed for three rounds of MDA (153/1000), for two rounds (13/1000), but not for one round. Note that there is no re-importation of malaria in this baseline scenario, and there is little to no pre-existing artemisinin resistance (no 580Y alleles) at the start of the simulation. During treatment, parasites can mutate from one genotype to another as long as the new genotype provides a

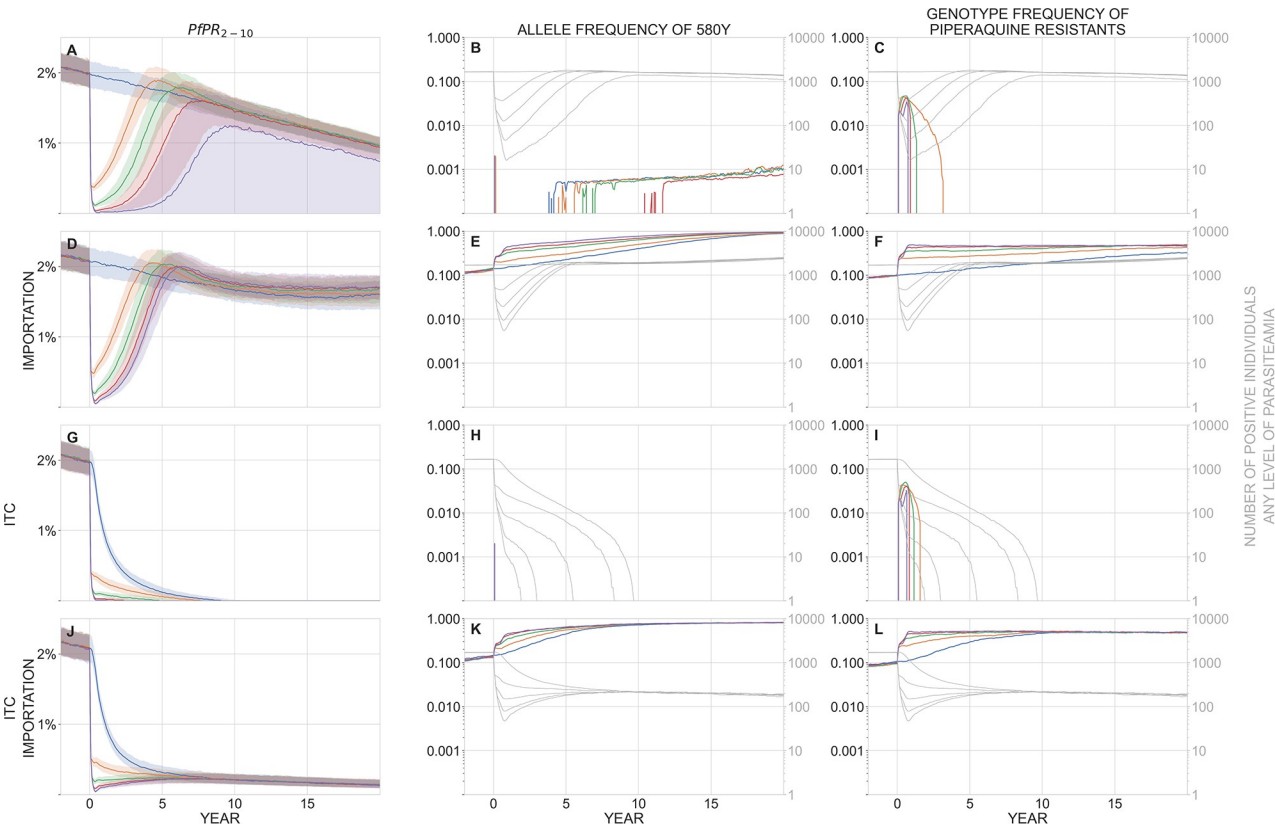

**Fig 2.** In a population of 40,000 individuals, panels show malaria prevalence (PfPR$_{2\text{-}10}$, left), allele frequency of 580Y (middle), and the genotype frequency of piperaquine-resistant parasites (right) over a period of 20 years after a mass drug administration has been carried out. In these scenarios, baseline PfPR$_{2\text{-}10}$ = 2%. Median trajectories are shown from 1000 simulations, and the shaded areas (left panels) show the interquartile range. Simulations are colored by the number of rounds of MDA carried out: blue (0), orange (1), green (2), red (3), purple (4). The top row (panels **A, B, C**) shows a scenario with no importation of drug-resistant genotypes and no improvement in treatment coverage (ITC) after the MDA is carried out. The second row (**D, E, F**) shows a scenario where new parasite importation occurs every 10 days; the imported parasite has a 50% probability of carrying the 580Y allele and a 50% probability of carrying piperaquine resistance, independently. The third row (**G, H, I**) shows a scenario with no importation, but where treatment coverage is increased post-MDA to 80% of the symptomatic patient population. The fourth row (**J, K, L**) shows a scenario with both importation and ITC. In the middle and right columns, the light gray lines show the absolute number of infected individuals (of any parasitaemia level) in the simulation and correspond to the right-hand gray tick marks on each panel. Note for example that in panels **K** and **L**, artemisinin-resistant and piperaquine-resistant genotype frequencies are very high, but in a population of only about 100 infected individuals, most of which are imports that occur during the course of the simulation; this region appears to have eliminated malaria but recently imported parasite-positive cases can still be found. MDA begins at year zero, and the first-year drop in prevalence can be seen in all plots in the left column. The bottleneck period lasts months to years, depending on the number of rounds of MDA carried out. Rapid selection of 580Y can be seen during the bottleneck period when importation is present (panels **E** and **K**). Piperaquine-resistant genotypes are maintained in the population through linkage disequilibrium with 580Y genotypes (panels **F** and **L**, see Fig F in S1 Text). The bottleneck period is risky when importation of 580Y alleles is expected; under these conditions, more rounds of MDA result in worse long-term drug-resistance outcomes (panel **E**).

fitness advantage in the presence of drug treatment (e.g. C580 to 580Y mutations are allowed during AL and DHA-PPQ treatment); however, both real-world and modeled *de novo* mutation rates are low enough that new mutants are rarely observed in a population of thousands of infected individuals pre-MDA or hundreds of infected individuals post-MDA. For this reason, artemisinin resistance takes a long time to emerge in a population of 40,000 individuals, and median 580Y allele frequencies in this scenario remain below 0.01 for the 20-year duration of the simulation, irrespective of the number of MDA rounds applied. The size and duration of the post-MDA bottleneck makes artemisinin-resistance emergence less likely under more rounds of MDA (Kruskal-Wallis $p < 10^{-91}$, $N = 1000$ runs, 580Y frequency taken at year 5, Fig 2B). Median 580Y allele frequency at year five is $<10^{-3}$ (it is zero for two or more rounds). Out

of $N = 1000$ simulations, the 90th percentile 580Y frequency at year five is 0.022 (no MDA), 0.009 (one round), 0.004 (two rounds), 0.002 (three rounds), and 0.001 (four rounds).

*P. falciparum* evolutionary dynamics following mass drug administration are qualitatively different when importation or re-importation of drug-resistant alleles is considered (Fig 2D to 2F), as would occur in a scenario where geographical areas receiving MDA are surrounded by untreated areas with drug resistance. When importation is occurring, the post-MDA bottleneck in the parasite population allows for more efficient selection of drug-resistant alleles: the smaller the bottleneck the faster the initial increase in gene frequency when a resistant allele is introduced. As expected, applying more rounds of MDA results in a smaller bottleneck and faster drug-resistance evolution. Fig 2E shows that each additional round of MDA has a detrimental effect on future artemisinin resistance when importation is allowed (KW $p < 10^{-188}$). The long-term driver of this effect is the continued post-MDA use of ACT as first-line therapy, which maintains selection pressure on 580Y throughout the entire post-MDA period. With no MDA, 580Y alleles reach a median frequency of 0.23 five years after MDA; if MDA is implemented, median 580Y frequencies five years later are 0.30 (one round), 0.43 (two rounds), 0.51 (three rounds), and 0.58 (four rounds).

At higher malaria prevalence and higher population size, the bottlenecking effect is weaker. In a population of 300,000 individuals with PfPR$_{2-10}$ = 5% (Fig 3), the parasite bottlenecks are larger (i.e. more parasite-positive individuals post-MDA) and of shorter duration, and elimination is not possible with ~80% MDA coverage; Figs A and B in S1 Text show these dynamics when only prevalence or only population size are changed, respectively. With no importation (Fig 3A to 3C), MDA has little effect on long-term artemisinin resistance evolution. After five years, median 580Y frequencies are between 0.002 and 0.007 depending on the number of MDA rounds applied. The effects of mutation in this scenario are small but not completely negligible since the population of infected individuals is large enough for 580Y mutations to have appeared at low frequencies (median 0.0031; IQR: 0.0011–0.0075) prior to the initiation of MDA. In other words, there is some genetic variation for the MDA to act on, which is why a small spike in 580Y frequencies is observed during the first year of the simulation. However, in the majority of scenarios, these mutants are rare and it is possible for them to not survive the bottlenecking period despite being selected for. With no importation, median 580Y frequencies (at year five) tend to go down with more rounds of MDA (KW $p = 10^{-52}$) with allele frequencies of 0.0066 (no MDA), 0.0071 (one round), 0.0060 (two rounds), 0.0032 (three rounds), and 0.0022 (four rounds). However, the magnitude of this effect is moderate-to-weak and the significant KW $p$-value is due to the large number of simulations. When re-importation is allowed (Fig 3D to 3F), we observe the same effects as in Fig 2D to 2F, namely that more rounds of MDA results in higher artemisinin-resistance frequencies in the long run (KW $p < 10^{-229}$). Piperaquine resistance is maintained in the population via positive linkage disequilibrium with artemisinin resistance (Figs F and G in S1 Text).

The major MDA-related risk to safeguard against is exacerbation of drug resistance, wherein MDA pushes the parasite population through a resistance bottleneck and post-MDA ACT use efficiently selects for drug-resistant alleles that the bottleneck recently bumped up to high frequency. In our modeling analysis, the greatest risk of this 'perverse effect' occurs when all three of the following elements are present: (1) a small genetic bottleneck post-MDA without eliminating the parasites, (2) importation or pre-existence of drug-resistant parasites, and (3) persistent selection for drug-resistant alleles post-MDA (Box 1). When no bottleneck is present, there are no rapid increases of gene frequency caused by small population sizes (compare blue and purple lines in Figs 2E and 3E). Without importation, there are no drug-resistant alleles to be selected for during the bottleneck and mutation is not rapid enough to generate them *de novo* when absolute population sizes are small (compare panels in Figs 2B/2E, 3B/3E). When there is no selection for artemisinin resistance or piperaquine resistance after the MDA,

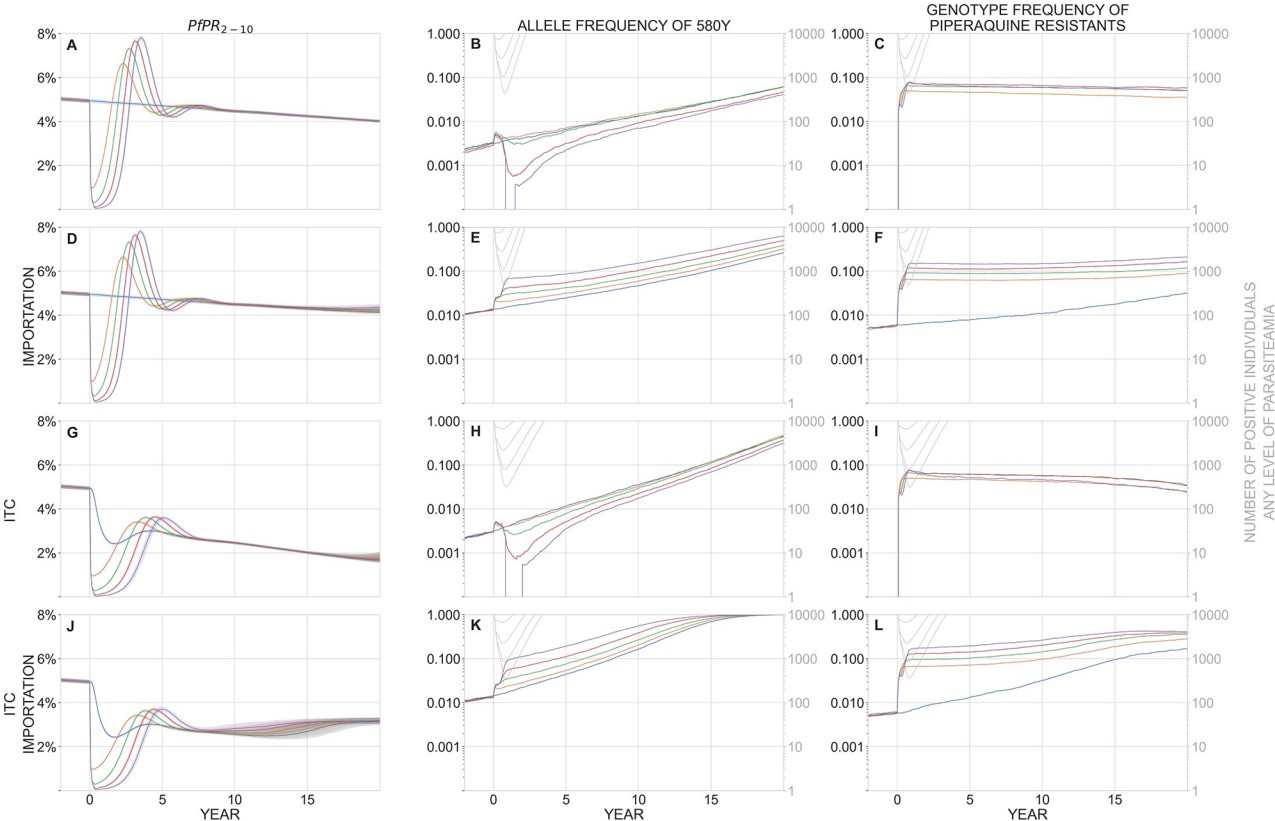

**Fig 3.** In a population of 300,000 individuals, panels show malaria prevalence (PfPR$_{2-10}$, left), allele frequency of 580Y (middle), and the genotype frequency of piperaquine-resistant parasites (right) over a period of 20 years after a mass drug administration has been carried out. In these scenarios, baseline PfPR$_{2-10}$ = 5%. Median trajectories are shown from 1000 simulations, and the shaded areas (left panels) show the interquartile range. Simulations are colored by the number of rounds of MDA carried out: blue (0), orange (1), green (2), red (3), purple (4). The top row (panels **A, B, C**) shows a scenario with no importation of drug-resistant genotypes and no improvement in treatment coverage (ITC) after the MDA is carried out. The second row (**D, E, F**) shows a scenario where new parasite importation occurs every 10 days; the imported parasite has a 50% probability of carrying the 580Y allele and a 50% probability of carrying piperaquine resistance, independently. The third row (**G, H, I**) shows a scenario with no importation, but where treatment coverage is increased post-MDA to 80% of the symptomatic patient population. The fourth row (**J, K, L**) shows a scenario with both importation and ITC. In the middle and right columns, the light gray lines show the absolute number of infected individuals (of any parasitaemia level) in the simulation and correspond to the right-hand gray tick marks on each panel. MDA begins at year zero, and the first-year drop in prevalence can be seen in all plots in the left column. The bottleneck period lasts months to years, depending on the number of rounds of MDA carried out. Selection of 580Y can be seen during the bottleneck period when importation is present (panels **E** and **K**), although this bottleneck effect is weaker in a population of 300,000 individuals (this figure) than a population of 40,000 individuals (Fig 2E and 2K). Piperaquine-resistant genotypes are maintained in the population through linkage disequilibrium with 580Y genotypes (panels **C, F, I, L**, see Fig G in S1 Text). The bottleneck period is risky when importation of 580Y alleles is expected; under these conditions, more rounds of MDA result in worse long-term drug-resistance outcomes (panels **E** and **K**).

these alleles recede from the population due to an assumed fitness cost for resistant genotypes. To verify this via simulation, we substituted in an OZ439-ferroquine combination [43–45] as first-line therapy after the MDA and showed that 580Y alleles are predicted to decrease in frequency after the MDA has completed (see Figs C to E in S1 Text). Therefore, if one of these three elements is missing, MDA is not predicted to exacerbate the population's drug-resistance levels as compared to a policy of no MDA; in these cases, MDA should have a neutral effect on drug-resistant alleles, and high-coverage MDA can be pursued without concern that drug-resistance will undermine the intended public health goals. Note that an MDA implementation followed by a small bottleneck and importation risks (elements 1 and 2 together) will still generate high levels of drug resistance, but this will not lead to treatment failures in patients if first-line therapy is changed after the MDA.

### Box 1

| | |
|---|---|
| **Recommendation 1** | In some scenarios, it may be possible that mass drug administration will drive the evolution of drug resistance more strongly when compared to no MDA. This risk exists when all three of the following are present after the MDA program: **(A)** a very small number of parasite-positive individuals, **(B)** regular importation of drug-resistant genotypes in the focal MDA area, and **(C)** continued use of first-line therapy that selects for these newly imported drug-resistant genotypes. |
| **Recommendation 2** | In the present modeling analysis, detrimental drug-resistance outcomes–if they occur–typically appear several years after the MDA. If a risk of higher drug resistance is identified, a course correction should be planned as soon as possible. |
| **Recommendation 3** | If one of **A**, **B**, or **C** (from **Recommendation 1**) is missing, MDA is projected to have a neutral or mitigating effect on drug-resistant alleles, and high-coverage MDA can be pursued without concern that drug-resistance will undermine the intended public health goals. |
| **Recommendation 4** | The regional epidemiological and genetic characteristics of *P. falciparum* transmission should be well understood before initiating MDA activities. If the MDA is targeted at a group of highly connected population, all of which have high drug-resistance levels, an MDA should be conducted for the entire region or not at all. If MDA implementation is only possible in some regions among a highly connected group, the regions with highest resistance levels should be chosen for MDA assuming prevalence levels are sufficiently low. |
| **Recommendation 5** | According to our modeling analysis, the level of MDA coverage has a weak effect on drug-resistance evolution. Therefore investment in community outreach, public communication, and staffing to lift MDA participation rates to the highest possible levels would likely increase the chances of local elimination without risking adverse drug-resistance outcomes. |
| **Recommendation 6** | The most direct approach to avoiding drug-resistance risks after several rounds of MDA is to drive parasite numbers to zero while the number of parasite-carrying individuals is still low. An economic analysis based on the allowable costs and projected benefits of post-MDA improvements in case management could help make the case that these additional interventions are potentially highly cost effective. |
| **Recommendation 7** | After an MDA program has completed, future malaria prevalence is more predictable than future drug-resistance evolution. The low-prevalence period after an MDA is an opportune period to drive parasite numbers to zero, and additional public health resources should be committed to this activity. |

The dangers of drug-resistance importation during MDA can be seen more clearly when we look at a classic 'waiting time' measure, here, the time until a critical and potentially dangerous level of drug resistance has been reached in the population. Choosing 25% artemisinin resistance as our milestone, Fig 4 shows that 580Y alleles increase in frequency and reach 0.25 allele frequency earlier when importation is more frequent. This effect is more pronounced at smaller population size (40,000 individuals) and more rounds of MDA, as it is in these smaller bottlenecks that imported drug-resistance alleles have the largest advantage by starting their fixation process at a higher frequency.

## 3.2 Improved treatment coverage (ITC) after the MDA

After an MDA program has completed all rounds, an opportunity presents itself to extinguish remaining chains of malaria transmission and achieve regional elimination. In September 2018, a WHO-convened evidence review group concluded that "maintaining reductions in transmission after the last round of MDA requires additional interventions, including vector control, case management, and intensified surveillance and response" [18]. These recommendations were reviewed by the Malaria Policy Advisory Committee (MPAC) in April 2019 emphasizing "that MDA must be thought of as a package together with other interventions" with a goal of reducing "transmission to the point that intensive case- and focus-based

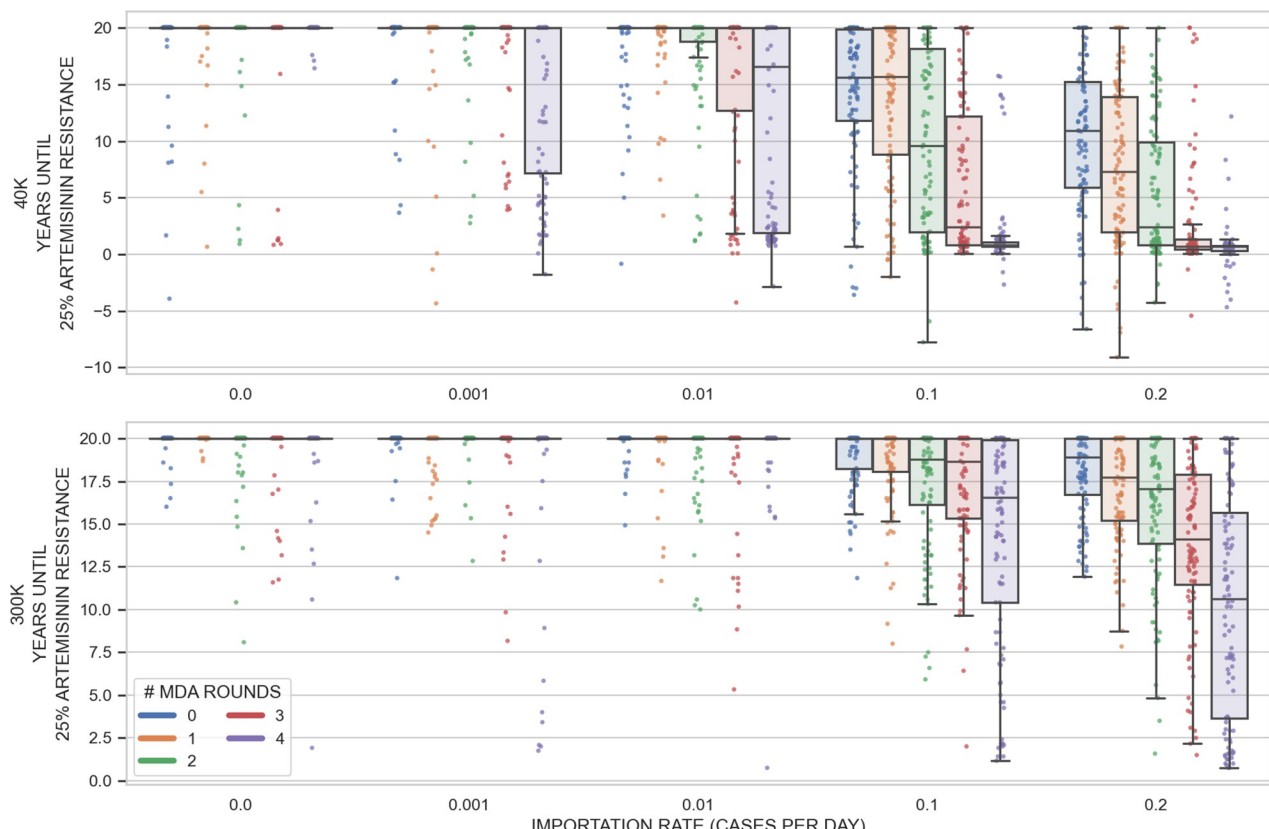

**Fig 4. Effect of importation on artemisinin-resistance evolution in an MDA scenario with PfPR$_{2-10}$ = 2%.** The box-and-scatter plots show how long it takes for the allele frequency of 580Y to reach 0.25 in the simulation. Note that both the drug used in MDA (DHA-PPQ) and the first-line drug used to treat symptomatic *P. falciparum* malaria (AL) contain an artemisinin derivative, so selection on 580Y is occurring throughout the simulation. Five importation rates were considered (*x*-axis). The imported genotype has an equal 25% probability of being any of the four genotypes C580/580Y and piperaquine sensitive/resistant. One hundred simulations were run for each importation rate, each of the two population sizes, and each possibility of zero to four rounds of MDA (5000 simulations in all). One dot corresponds to one simulation and the boxes are interquartile ranges. With no or very low importation (0.001/day), there is little genetic variation in the population and the bottleneck period typically has no effect on selection. When importation of drug-resistance is common, it is likely that drug-resistant genotypes will be present during the bottlenecking process; in these scenarios, more rounds or MDA lead to smaller bottlenecks and smaller bottlenecks result in more rapid selection. Treatment coverage is 55% for individuals over the age of 5, and 60% for children <5, and there is no improvement of treatment coverage in the post-MDA period. Negative values are present in the top panel as high rates of importation can lead to >0.25 580Y allele frequency before the initiation of MDA. Note that the simulation is run for twenty years, so any points at the top of the graph simply mean twenty or more years.

activities can be initiated" [46]. Field studies have confirmed that improved treatment coverage both pre- and post-MDA is likely to increase the chances of elimination and reduce the probability of an epidemiological rebound [3, 33].

In a scenario with no re-importation of malaria, our model shows that improved treatment coverage (ITC) post-MDA can lead to malaria elimination if prevalence is low enough (Fig 2G to 2I) but not if the starting prevalence is high (Fig 3G to 3I). When re-importation of malaria cases is expected, a post-MDA ITC policy can still lead to elimination. Elimination in these settings is defined as a scenario where malaria may be imported but will not generate enough secondary cases to initiate an outbreak or establish endemically; imported cases in this scenario are eventually cured and/or treated as are the small number of secondary cases they generate. Fig 5 shows prevalence reductions five years post-MDA in a setting with PfPR$_{2-10}$ = 1% and frequent re-importation. In a population of 40,000 individuals, four rounds of MDA and no ITC leads to a majority of simulations (57%) having fewer than 100 infected individuals (practically speaking, an elimination outcome), but 94% and 98% of simulations reach the <100

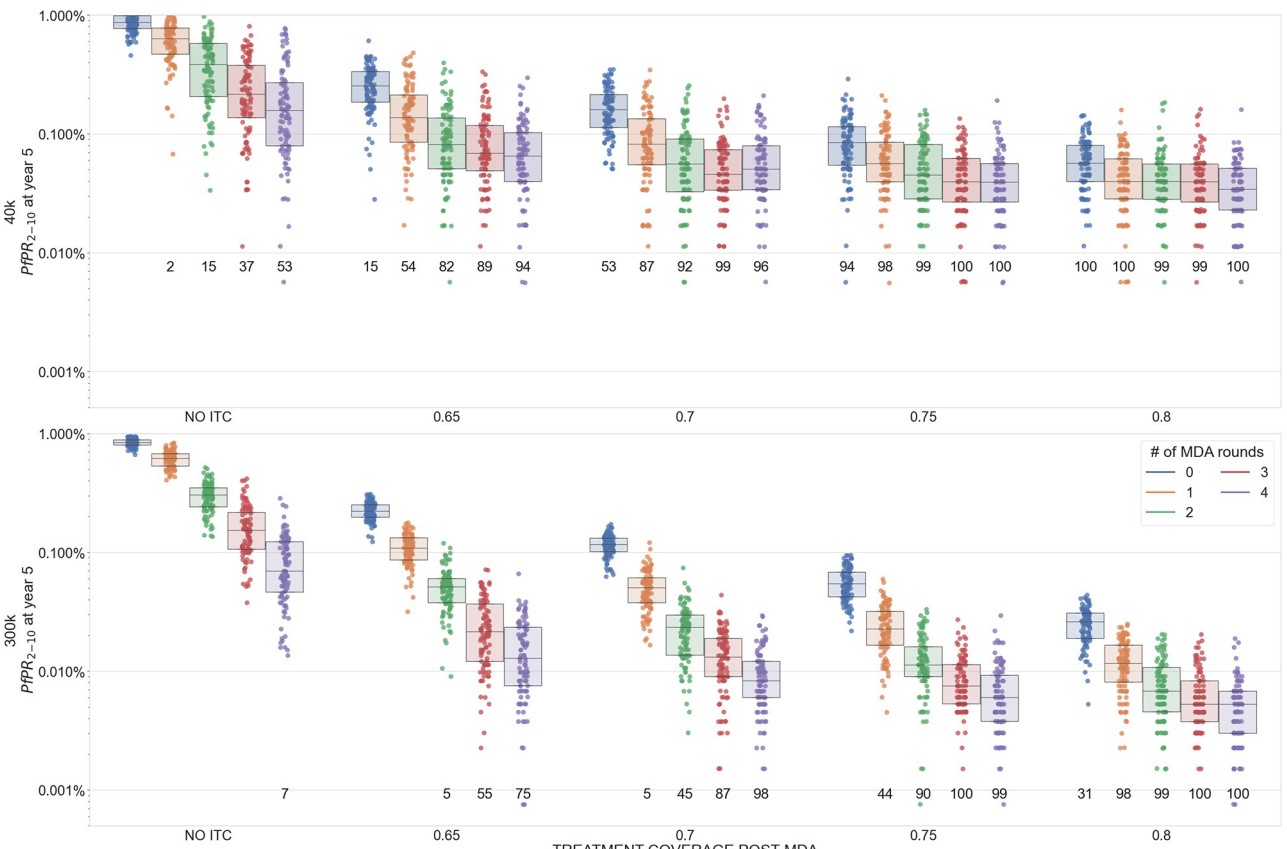

**Fig 5. Malaria prevalence five years after MDA (box plots summarize 100 simulations each) with baseline PfPR$_{2-10}$ = 1% and importation of 580Y alleles set at 0.1 cases/day.** The numbers printed below each plot indicate the number of simulations that reached fewer than 100 malaria cases present in the population (i.e. almost all are imports), which is a proxy for the number of simulations that achieved elimination. Numbers of rounds of MDA are shown by different colors, and four levels of improved treatment access are considered on the *x*-axis (65% coverage post-MDA to 80% coverage post-MDA). Improved treatment coverage post-MDA increases the chances of local elimination.

infections point when antimalarial treatment coverage post-MDA reaches 65% and 70%, respectively. In a population of 300,000 individuals, the probability of reaching elimination improves from 15% (no ITC) to 85% (ITC at 65%) and 98% (ITC at 70%) when improving access and coverage of antimalarials after the MDA. As long as baseline prevalence is low enough, MDA+ITC can lead to elimination even under a scenario of constant re-importation of drug-resistant malaria.

When baseline prevalence is high (PfPR$_{2-10}$ ≥ 5%), malaria infections will rebound post MDA, even when ITC is made available (Fig 3, 3K, and 3L). When re-importation is likely, this rebound will carry with it a high frequency of drug resistance (higher than if no MDA had taken place), and this is the countervailing evolutionary-epidemiological effect that must be avoided when considering MDA in higher prevalence settings (unless MDA participation and/ or ITC levels can be pushed to higher levels than are modeled in the present analysis).

### 3.3 Variation and predictability

In general, the bottleneck dynamics in the simulation result in a wide range of variability in simulation outcomes, with allele frequency trajectories having low predictability and prevalence trajectories having higher predictability. Runs of one thousand simulations per scenario allow us to explore this range of outcomes from the 1st to the 99th quantiles. At 5% prevalence,

allele frequencies of artemisinin resistance are not predictable post-MDA, with the middle 98% range of outcomes generally spanning two to three orders of magnitude (Figs R and T in S1 Text). At 2% prevalence with no importation, artemisinin-resistance frequencies are somewhat more predictable as the chances of elimination are high and there are no imported resistant alleles for selection to operate on (panel B, H in Fig Q; and panel H in Fig S in S1 Text). However, at 2% prevalence with resistance importation, natural selection acts with variable efficiency depending on how small the population of infections is; the result is that 580Y allele frequencies can range from a frequency of 0.02 to near fixation five years after the MDA has completed (panel E, K in Fig Q, and panel E in Fig S in S1 Text).

### 3.4. Sensitivity analysis

A sensitivity analysis on potentially influential epidemiological and evolutionary parameters–all controllable with the exception of the drug-resistant genotypes' fitness cost–shows that pre-MDA malaria prevalence has the highest correlation (Kendall PRCC = +0.45) with the time until artemisinin resistance reaches 0.25 genotype frequency ($T_{0.25}$); see Fig 6. This means that

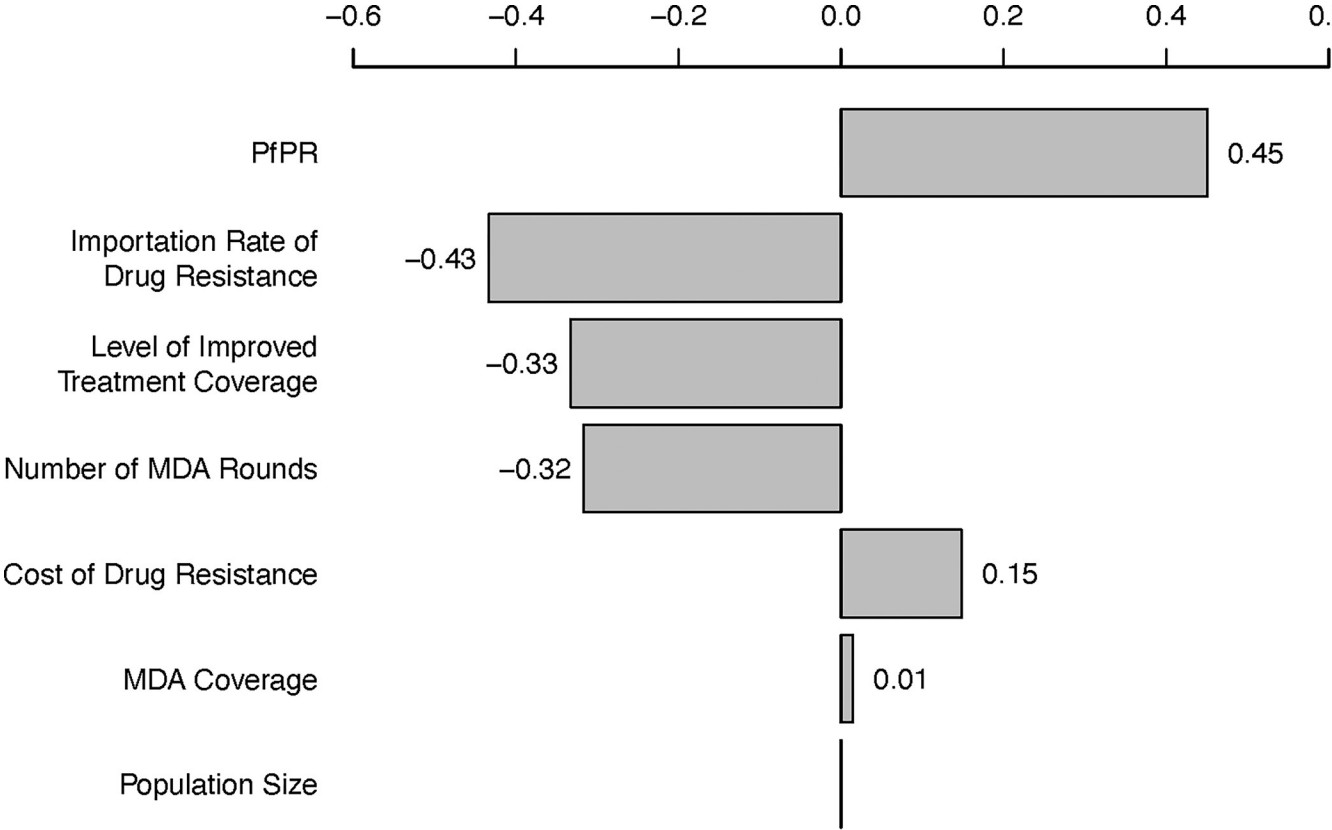

**Fig 6. Tornado plot of Partial Rank Correlation Coefficient (PRCC) between various epidemiological variables (left column) and the time until artemisinin-resistance allele frequency reaches 0.25 ($T_{.25}$).** $T_{.25}$ is found to be positively correlated with PfPR ($r = 0.45$), and negatively correlated with importation rate ($r = -0.43$), the post-MDA level of improved treatment coverage ($r = -0.33$), and the number of deployed MDA rounds ($r = -0.32$). Fitness cost of drug resistance has a moderate positive correlation with $T_{.25}$ ($r = -0.15$). MDA coverage and population size are not associated with slower or faster artemisinin-resistance evolution ($r = 0.01$, $p = 0.002$ and $r = 0$, $p = 0.9$, respectively). All $p$-values are lower than $10^{-5}$ except where indicated.

although high prevalence settings are the least likely to reach elimination, they do exhibit the slowest artemisinin-resistance evolution, directly showing that these two opposing effects of prevalence must be considered in both MDA and post-MDA planning. The importation rate of drug-resistant genotypes has the second strongest correlation with $T_{0.25}$ (PRCC = -0.43), a likely consequence of the dominant role importation plays in introducing drug-resistant genotypes and the small role that *de novo* mutation plays in the short-term bottlenecked population genetics of mass drug administration. The number of MDA rounds carried out (PRCC = -0.33) and the post-MDA ITC (PRCC = -0.32) have the next largest correlations with artemisinin-resistance evolution, again showing that certain covariates that increase the probability of malaria elimination may increase long-term drug-resistance risk (if no corrective course is taken after the MDA, and if importation rates are high). The fitness cost of drug-resistance (PRCC = +0.15) has the expected effect on drug-resistance evolution, but these parameters are rarely measured in the field and are not controllable. The overall MDA coverage (PRCC = +0.01) and focal population size (PRCC = 0.00) are not associated with any notable differences in $T_{0.25}$. The partial rank correlation between the bottleneck size and future 580Y frequency is between -0.31 and -0.27; see S1 text, section 5.

Several of these epidemiological features–pre-MDA prevalence, number of MDA rounds, post-MDA ITC–are not manipulable without trading off future drug-resistance risk against current MDA success. Lower prevalence, more rounds of MDA, and higher ITC are simultaneously associated with (*i*) increased probability of near-term malaria elimination and (*ii*) higher 580Y frequencies in the long term. For two epidemiological features however, no such trade-off exists. First, high MDA coverage with DHA-PPQ is not associated with higher artemisinin resistance. Our analysis shows that the increased short-term selection pressure generated during MDA is not the cause of detrimental resistance outcomes far into the future; rather, it is the continual drug pressure post-MDA that is the primary driver of subsequent resistance evolution (Fig C to Fig E in S1 Text). Second, lower importation risk of drug-resistant alleles is associated with slower artemisinin-resistance evolution post MDA, with no trade-offs or unexpected benefits of higher importation rates. This indicates that the regional genetic epidemiology of *P. falciparum* will be critical in defining candidate populations for MDA programs, for the sole purpose of ensuring that drug-resistant parasites cannot be re-introduced as malaria is approaching local elimination.

## 4 Discussion

This work was commissioned by the WHO Evidence Review Group on Mass Drug Administration for Malaria [18] and is intended to provide guidance for MDA scenarios in Africa–in particular, in locations where there may be importation risks of artemisinin-resistant genotypes [47, 48]–and higher-prevalence SE Asia scenarios where artemisinin resistance is common and widespread [49]. As artemisinin resistance has now emerged in many parts of the world [47–52], drug-resistance considerations should be taken into account for all planned MDA activities. Target population sizes of 40,000 and 300,000 individuals were evaluated as this was stated as the likely range of larger MDA programs supported by WHO [18]. In these settings, we find that strong population bottlenecks are associated with substantial uncertainty with respect to future patterns of antimalarial drug resistance, and we provide guidance on how to classify epidemiological scenarios when evaluating MDA-related drug-resistance risks. We discuss below some of the factors that are associated with the best possible outcome of an MDA, local malaria elimination, as well as the worst possible outcome, a rebound in malaria prevalence with higher drug-resistance levels than those expected under no MDA.

### 4.1 Conclusions

A combination of migration, drift, and selection–when all three are present simultaneously–can sometimes lead to a paradoxical epidemiological outcome where an MDA worsens long-term population-level health outcomes (Figs 2E, 3E, and 3K). In an epidemiological scenario with (1) a small bottleneck caused by the MDA, (2) likely re-importation of drug-resistant parasites, and (3) persistent selection pressure by the recommended first-line antimalarial post-MDA, long-term dynamics post-MDA can lead to an epidemiological rebound and higher frequencies of drug-resistance alleles when compared to a scenario with no MDA. Note that these detrimental outcomes typically appear several years after the MDA, allowing for a course correction if the risk has been identified. In addition, the re-importation requirement means that one would need to identify two highly connected populations with substantial drug resistance and perform MDA in only one of them to generate a worse-than-status-quo drug-resistance outcome. This serves as an obvious reminder to scrutinize regional epidemiological patterns before initiating MDA activities [3, 33], and to focus MDA on higher-resistance areas first if prevalence levels there are sufficiently low (see Box 1).

In many African malaria settings conducive to an MDA-based prevalence reduction, importation of artemisinin-resistance alleles is entirely dependent on the location chosen. The three requirements above will be of special importance to many local malaria management questions in Southeast Asia as some areas will meet all three conditions. In a Southeast Asian setting with high local prevalence and a high chance of re-importation of artemisinin resistance, mass drug administration may exacerbate long-term drug resistance risks if corrective action is not taken after the MDA. The three criteria above help us underline the obvious conclusion that in a group of connected populations all of which have high drug-resistance levels, an MDA should be conducted for the entire region or not at all. A partial MDA implementation in this scenario would likely cause re-imported drug resistance to nullify the near-term beneficial effects of MDA. Fortunately, the two SE Asian MDA projects to date that have recorded *Pfkelch13* allele frequencies to assess artemisinin-resistance risk did not detect an increase in resistance post-MDA [3, 11]. We propose that these criteria be evaluated in future considerations of MDA in large populations. Criterion 1 (small bottleneck) will likely be met by any successful MDA program. The focus would thus be on questions of drug-resistance importation and the ability to substitute out the first-line therapy once the MDA is complete.

Identification of these three conditions and their relationship to drug-resistance risk clarifies that the primary evolutionary force driving drug resistance is the first-line therapy used for routine case management post-MDA and not the MDA drug itself (Figs C to E in S1 Text), as the post-MDA bottleneck period is much longer than the actual MDA implementation. This presents a new policy option of substituting out the first-line therapy after an MDA has completed, in order to remove selection pressure on any recent large jumps in drug resistance. This policy option will need to be evaluated using specific geographic and treatment scenarios with a simulation approach similar to the one taken here. The principle that diverse drug environments will slow the evolutionary process should apply to this situation. The more treatment heterogeneity we can introduce into the parasites' environment during and post-MDA, the more difficult it will be for newly emergent drug-resistant genotypes to spread successfully [53–57].

The most direct approach to avoiding drug-resistance risks that materialize during or after an MDA bottleneck is to drive parasite numbers to zero while the number of parasite-carrying individuals is still low. In our analysis, a key factor in reaching elimination is the treatment coverage post-MDA. If the infrastructure and knowledge put in place for the MDA–e.g. newly trained staff, larger catchment area for a local malaria post, increase in community health

workers, stable procurement of antimalarials, awareness among febrile individuals on how to access effective antimalarials–is able to be maintained in a way that improves treatment coverage in the population, elimination can be achieved in low-prevalence settings (Fig 5). The cost-effectiveness of post-MDA ITC needs to be assessed. As twelve months of a large-scale population health program are unlikely to cost twice as much as six months of the same program, an important question in the health economics of malaria management will be determining how long to run an ITC campaign for after an MDA has completed.

If a regional or national health authority is concerned that it will not be possible to drive infection numbers to zero post-MDA, it will be crucial to review the controllable elements of the given epidemiological scenario and whether these elements involve a trade-off between low case numbers and low resistance levels. Among these, the level of MDA coverage has the weakest effect on drug-resistance evolution (Fig 6) suggesting that investment in community outreach, public communication, and staffing to lift MDA participation rates to the highest possible levels would be the most prudent use of time and resources.

## 4.2. Limitations

One current limitation of the modeling analyses presented here is the generalizability of these scenarios to all malaria-endemic settings. Many important epidemiological characteristics such as seasonal changes in mosquito populations, geographic extent of the MDA program [58], variation in household biting rates, and participation and compliance in each MDA round [14] will have important effects on the success of MDA programs. While it is true that African and Southeast Asian regions will have markedly different risks corresponding to different resistance profiles of circulating *P. falciparum* populations, partner-drug resistance will be present in both settings and will need to be accounted for as a potential cause of current and future treatment failures [20]. As elimination plans move forward regionally in SE Asia, as well as elimination-ready regions in Africa and South America, MDA model scenarios like the ones outlined here will need to be tailored to the exact populations and epidemiological characteristics of regions whose prevalence has been reduced to the point where MDA programs can realistically be used to accelerate towards elimination.

A second major limitation in forecasting *P. falciparum* drug-resistance evolution of any kind is the large number of loci, epistatic interactions, and genotype-specific treatment efficacies that need to be imputed or estimated to allow a model to generate a prediction of which genotypes will be selected more strongly than others. The present analysis uses six loci (with a seventh locus added in Figs C to E in S1 Text) which presents a computational challenge as a recombination table for $n$ loci requires $2^{3n}$ entries, an approach that is unsustainable for large numbers of loci. Crucially, piperaquine resistance appears to be affected by at least five candidate loci in the *pfcrt* gene [26, 29, 30, 59] which presents an urgent need for (*i*) a computationally more efficient recombination mechanisms that can accommodate 15 or more loci, and (*ii*) further literature review and parameter estimation on the individual effects that the alleles at these loci have on 28-day treatment efficacy. An additional benefit of expanding the number of loci will be increased model certainty on the effect of private-market drug use, which can generate substantial selection pressure and alter long-term evolutionary outcomes [21]. In the present analysis, SP resistance levels are fixed (but see Figs I to L in S1 Text) but a more complete model of the *P. falciparum* genotype-to-phenotype drug-resistance map would improve predictions of drug-resistance evolution driven by both public-sector distribution and private-sector drug sales.

During and after a mass drug administration program, prevalence management strategies yield more programmatic certainty than resistance management strategies. Low parasite

numbers immediately post-MDA present an opportune period to drive prevalence to zero, but they do not provide good predictability on near-term evolutionary outcomes related to drug resistance. As more malaria-endemic regions are brought to low-prevalence status, MDA will become more called upon as a policy option. Success will depend on assessment and management of re-importation risk, ability of health systems to commit resources to ensure high participation, long follow-up, and continual improvements in malaria control after the MDA program has finished.

## Supporting information

**S1 Text. Supplemental materials 1.** This PDF file contains information concerning the model parameterization, calibration, validation, and additional sensitivity analyses.
(PDF)

**S2 Text. Supplemental materials 2.** This PDF file contains additional information that describes the process of creating a comprehensive table approximating the effectiveness of various antimalarial treatments on sixty-four different *P. falciparum* genotypes, facilitating the development of mathematical models and simulations and therapeutic effects on any genotype.
(PDF)

## Acknowledgments

Thanks to Lorenz von Seidlein and François Nosten for valuable discussions.

## Author Contributions

**Conceptualization:** Nicholas J. White, Maciej F. Boni.

**Data curation:** Thu Nguyen-Anh Tran, Daniel M. Parker.

**Formal analysis:** Tran Dang Nguyen, Thu Nguyen-Anh Tran, Daniel M. Parker, Maciej F. Boni.

**Funding acquisition:** Maciej F. Boni.

**Investigation:** Tran Dang Nguyen, Thu Nguyen-Anh Tran, Maciej F. Boni.

**Methodology:** Tran Dang Nguyen, Thu Nguyen-Anh Tran, Maciej F. Boni.

**Project administration:** Maciej F. Boni.

**Supervision:** Nicholas J. White, Maciej F. Boni.

**Validation:** Tran Dang Nguyen, Thu Nguyen-Anh Tran, Daniel M. Parker, Maciej F. Boni.

**Writing – original draft:** Maciej F. Boni.

**Writing – review & editing:** Tran Dang Nguyen, Daniel M. Parker, Nicholas J. White, Maciej F. Boni.

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
