## [Decision Letter · Decision Letter 0]

3 May 2023

PGPH-D-23-00382

Antimalarial mass drug administration in large populations and the evolution of drug resistance

Dear Dr. Boni,

Thank you for submitting your manuscript to PLOS Global Public Health. After careful consideration, we feel that it has merit but does not fully meet PLOS Global Public Health’s publication criteria as it currently stands. Therefore, we invite you to submit a revised version of the manuscript that addresses the points raised during the review process.

We look forward to receiving your revised manuscript.

Kind regards,

André Machado Siqueira, M.D., MSc, Ph.D

Academic Editor

Journal Requirements:

1. We ask that a manuscript source file is provided at Revision. Please upload your manuscript file as a .doc, .docx, .rtf or .tex.

Additional Editor Comments (if provided):

Reviewers' comments:

Reviewer's Responses to Questions

**Comments to the Author**

1. Does this manuscript meet PLOS Global Public Health’s publication criteria? Is the manuscript technically sound, and do the data support the conclusions? The manuscript must describe methodologically and ethically rigorous research with conclusions that are appropriately drawn based on the data presented.

Reviewer #1: Yes

Reviewer #2: Partly

2. Has the statistical analysis been performed appropriately and rigorously?

Reviewer #1: I don't know

Reviewer #2: No

3. Have the authors made all data underlying the findings in their manuscript fully available (please refer to the Data Availability Statement at the start of the manuscript PDF file)?

Reviewer #1: Yes

Reviewer #2: Yes

4. Is the manuscript presented in an intelligible fashion and written in standard English?

Reviewer #1: Yes

Reviewer #2: Yes

5. Review Comments to the Author

Reviewer #1: This study raises a critical issue to consider when implementing MDA in different context. The manuscript is well written with comprehensive review of relevant evidences for constructing assumptions and model inputs. The conclusions and recommendations are important for public health intervention program to carefully plan to achieve the optimal outcomes to avoid future negative impact.

A minor suggestion: If the authors could provide few bullet points to summarize the most important findings, following by key recommendations related to MDA implementation, in a simple language that non-math people could understand, the readers will help accelerate the findings further, such as to policy makers.

Reviewer #2: The goal of this study is timely for understanding the effect of MDA coverage on the emergence of antimalarial resistance. The proposed scenarios are well-designed and can lead to interesting conclusions, particularly regarding the time it takes for resistance to arise under specific conditions, such as importation of cases. However, a few issues require attention in the model, description of results, and discussion. In particular, a higher number of simulations would better support the results. The current number of 100 simulations for each combination of parameters is low, which may impact the validity of some of the results and curves in the figures. While the conclusions may not change dramatically, it is important to clearly indicate the limitations and more simulations are required.

I provide some specific (major and minor) comments on the manuscript as follows.

Abstract

The sentence in line 5 (“Effective MDA5… ” ) can be misleading since utilization of MDA might impose selective pressure. This is in fact acknowledged in the next sentences. I suggest revision.

Introduction

Line 40 - I recommend saying if genotypes are from P. falciparum, P. vivax or both. The Introduction does not make much distinction between these two species.

Model

Describe what EC50 is.

Line 86: are these numbers arbitrary, such as 55% receiving treatment?

Again, importation every 10 days seems arbitrary.

Regarding equal probability of importation of genotypes, does it not make sense that this importation would follow the frequencies of the most common sources of importation?

Line 145 (section 2.3): this seems to be results reported in the methodology? If so, I suggest to have it in the next section.

Line 160 - 174: part of this material could appear in Introduction or Discussion.

Description of T0.25 is required.

How is the Fitness cost modeled?

How many simulations are run in each scenario?

Results

Line 192: elimination observed for three simulations (8/100), anything misreported here?

Line 266 - 275: sounds as Introduction or Discussion

Line 295 - 299: This seems more a material for the Discussion

Lines 330 - 332: critical for discussion

Figure 1: indicate the meaning of orange box and other boxes and colors. Need to find a more meaningful label than Num Pos Indivs. What scale is for the x-axis?

Number of simulations is indicated here as 100 for each combination of parameters: number is low.

For instance, in Figure 1 I expect that when plotting the average numbers the curves will be better behaved. The same is likely to happen to Figure 2 in the scenarios without importation.

Figure 3: for panels F and L, one of the curves is much lower. What is the explanation?

Figure 4: Is this figure indicating that for low levels of importation it will take 20 years to reach 25% of resistance? Since the maximum time in the simulations was 20 years, then it seems that 20 years appears here by an artifact of this simulation scenarios.

I would recommend highlighting the effects, in particular the negative effect in the sensitivity analysis for the importation rate and the MDA coverage.

Discussion

What do the authors mean with “drive parasite numbers to zero while the number of parasite-carrying individuals is still low”?

The sensitivity analysis deserves a discussion especially in terms of the negative effects that were found.

6. PLOS authors have the option to publish the peer review history of their article (what does this mean?). If published, this will include your full peer review and any attached files.

**Do you want your identity to be public for this peer review?** For information about this choice, including consent withdrawal, please see our Privacy Policy.

Reviewer #1: No

Reviewer #2: No

---

## [Decision Letter · Decision Letter 1]

3 Jul 2023

Antimalarial mass drug administration in large populations and the evolution of drug resistance

PGPH-D-23-00382R1

Dear Dr Boni,

We are pleased to inform you that your manuscript 'Antimalarial mass drug administration in large populations and the evolution of drug resistance' has been provisionally accepted for publication in PLOS Global Public Health.

Best regards,

André Machado Siqueira, M.D., MSc, Ph.D

Academic Editor

Reviewer Comments (if any, and for reference):

Reviewer's Responses to Questions

**Comments to the Author**

1. If the authors have adequately addressed your comments raised in a previous round of review and you feel that this manuscript is now acceptable for publication, you may indicate that here to bypass the “Comments to the Author” section, enter your conflict of interest statement in the “Confidential to Editor” section, and submit your "Accept" recommendation.

Reviewer #1: All comments have been addressed

Reviewer #2: All comments have been addressed

2. Does this manuscript meet PLOS Global Public Health’s publication criteria? Is the manuscript technically sound, and do the data support the conclusions? The manuscript must describe methodologically and ethically rigorous research with conclusions that are appropriately drawn based on the data presented.

Reviewer #1: Yes

Reviewer #2: Yes

3. Has the statistical analysis been performed appropriately and rigorously?

Reviewer #1: I don't know

Reviewer #2: Yes

4. Have the authors made all data underlying the findings in their manuscript fully available (please refer to the Data Availability Statement at the start of the manuscript PDF file)?

Reviewer #1: Yes

Reviewer #2: Yes

5. Is the manuscript presented in an intelligible fashion and written in standard English?

Reviewer #1: Yes

Reviewer #2: Yes

6. Review Comments to the Author

Reviewer #1: I have no further comment.

Reviewer #2: This revised manuscript improved significantly, in particular it was good to have the simulations with N=1000 runs, even though conclusions did not change, as I expected.

I liked very much the comments on the timing of resistance emergence.

It is great to be able to observe the outcomes from 1 to 99th percentiles.

The policy box panel was a great addition.

Also, the definition of the fitness cost and its model was important.

In page 6 I suggest rephrasing "from N=100 to N=1000", but this is just minor.

7. PLOS authors have the option to publish the peer review history of their article (what does this mean?). If published, this will include your full peer review and any attached files.

**Do you want your identity to be public for this peer review?** For information about this choice, including consent withdrawal, please see our Privacy Policy.

Reviewer #1: No

Reviewer #2: No
